# Food Intervention with Folate Reduces TNF-α and Interleukin Levels in Overweight and Obese Women with the *MTHFR* C677T Polymorphism: A Randomized Trial

**DOI:** 10.3390/nu12020361

**Published:** 2020-01-30

**Authors:** Jéssica Vanessa de Carvalho Lisboa, Marina Ramalho Ribeiro, Rafaella Cristhine Pordeus Luna, Raquel Patrícia Ataíde Lima, Rayner Anderson Ferreira do Nascimento, Mussara Gomes Cavalcante Alves Monteiro, Keylha Querino de Farias Lima, Carla Patrícia Novaes dos Santos Fechine, Naila Francis Paulo de Oliveira, Darlene Camati Persuhn, Robson Cavalcante Veras, Maria da Conceição Rodrigues Gonçalves, Flávia Emília Leite de Lima Ferreira, Roberto Teixeira Lima, Alexandre Sérgio da Silva, Alcides da Silva Diniz, Aléssio Tony Cavalcanti de Almeida, Ronei Marcos de Moraes, Eliseu Verly Junior, Maria José de Carvalho Costa

**Affiliations:** 1Postgraduate Program in Nutrition Sciences, Health Sciences Center, Federal University of Paraíba, João Pessoa 58059-900, Brazil; nutri.marinaramalho@gmail.com (M.R.R.); rafaellacpluna@gmail.com (R.C.P.L.); raquelpatriciaal@hotmail.com (R.P.A.L.); mussara.monteiro@hotmail.com (M.G.C.A.M.); k_farias1@hotmail.com (K.Q.d.F.L.); carlafechine@hotmail.com (C.P.N.d.S.F.); darlenecp@hotmail.com (D.C.P.); drrobveras@gmail.com (R.C.V.); mariadaconceicaorgoncalves@gmail.com (M.d.C.R.G.); flaemilia@gmail.com (F.E.L.d.L.F.); robertotexlima@gmail.com (R.T.L.); alexandresergiosilva@yahoo.com.br (A.S.d.S.); mjc.costa@terra.com.br (M.J.d.C.C.); 2Postgraduate Program in Molecular and Human Biology, Center of Exact and Natural Sciences, Federal University of Paraíba, João Pessoa 58059-900, Brazil; raynerbiomedicina@gmail.com; 3Departament of Molecular Biology, Federal University of Paraíba, João Pessoa 58059-900, Brazil; nailafpo@gmail.com; 4Postgraduate Program in Nutrition Sciences, Federal University of Pernambuco, Recife 50670901, Brazil; diniz.alcides@hotmail.com; 5Department of Economics, Postgraduate Program in App1lied Economics and Economics of the Public Sector, Center for Applied Social Sciences, Federal University of Paraíba, João Pessoa 58059-900, Brazil; alessiotony@gmail.com; 6Postgraduate Program in Health Decision Models, Federal University of Paraíba, João Pessoa 58059-900, Brazil; ronei@de.ufpb.br; 7Department of Epidemiology, Institute of Social Medicine, State University of Rio de Janeiro, Rio de Janeiro 20550-900, Brazil; eliseujunior@gmail.com

**Keywords:** interleukins, tumor necrosis factor-alpha, dietary intervention, polymorphism, genetic variants, folate, overweight, obesity

## Abstract

Methylenetetrahydrofolate reductase (*MTHFR*) C677T polymorphism associated with body fat accumulation could possibly trigger an inflammatory process by elevating homocysteine levels and increasing cytokine production, causing several diseases. This study aimed to evaluate the effects of food intervention, and not folate supplements, on the levels of tumor necrosis factor-α (TNF-α), interleukin-6 (IL-6), and interleukin-1β (IL-1β) in overweight and obese women with the *MTHFR* C677T polymorphism. A randomized, double-blind eight-week clinical trial of 48 overweight and obese women was conducted. Participants were randomly assigned into two groups. They received 300 g of vegetables daily for eight weeks containing different doses of folate: 95 µg/day for Group 1 and 191 µg/day for Group 2. *MTHFR* C677T polymorphism genotyping was assessed by digestion with HinfI enzyme and on 12% polyacrylamide gels. Anthropometric measurements, 24-h dietary recall, and biochemical analysis (blood folic acid, vitamin B12, homocysteine (Hcy), TNF-α, IL-1β, and IL-6) were determined at the beginning and end of the study. Group 2 had a significant increase in folate intake (*p* < 0.001) and plasma folic acid (*p* < 0.05) for individuals with the cytosine–cytosine (CC), cytosine–thymine (CT), and thymine–thymine (TT) genotypes. However, only individuals with the TT genotype presented reduced levels of Hcy, TNF-α, IL-6, and IL-1β (*p* < 0.001). Group 1 showed significant differences in folate consumption (*p* < 0.001) and folic acid levels (*p* < 0.05) for individuals with the CT and TT genotypes. Food intervention with folate from vegetables increased folic acid levels and reduced interleukins, TNF-α, and Hcy levels, mainly for individuals with the TT genotype.

## 1. Introduction

Women with an unbalanced diet in calories could have an excessive accumulation of body fat, especially in the abdominal circumference, and an elevated body mass index (BMI). This nutritional status of overweight and obesity generates systemic subclinical inflammation from the adipose tissue. In this situation, nuclear factor kappa-light-chain-enhancer (NF-κB) of the active B cell pathway increases cytokine synthesis. Thus, this inflammation increases the production of reactive oxygen species (ROS), triggering oxidative stress, and as a consequence, there is a higher production of cytokines, creating a vicious cycle. In addition, another factor contributing to oxidative stress is the high level of homocysteine (Hcy) in the blood. Hcy, a metabolite derived from folate metabolism, may be increased in the presence of methylenetetrahydrofolate reductase (*MTHFR*) C677T polymorphism [1,2,3].

The C677T polymorphism (rs1801133) consists in nonsynonymous substitution (alanine to valine exchange at the 222nd codon - Ala222Val) in the exon 4 of the *MTHFR* gene, and as a result, the enzyme methylenetetrahydrofolate reductase (*MTHFR*) produced is very thermolabile and with decreased activity [4]. Then, with reduced activity, this key enzyme in folate metabolism is not able to perfectly catalyze the 5,10-methylenetetrahydrofolate (MTHF) reduction reaction in 5-MTHF and donate a methyl (CH3) group to the Hcy remethylation pathway in methionine, leading to hyperhomocysteinemia [4,5,6]. The high level of this metabolite (Hcy) leads to oxidative stress and also contributes to the elevation of inflammatory biomarkers, such as tumor necrosis factor-α (TNF-α), interleukin-6 (IL-6), and interleukin-1β (IL-1β) [7,8,9,10,11,12]. Furthermore, high Hcy oxidation and body fat accumulation could lead to cytokine-mediated inflammation, and an increase in the oxidative stress. In addition, a balanced diet rich in antioxidants and anti-inflammatory nutrients like folate might be necessary to reduce this damage and also to avoid the C677T polymorphism effects [3,13,14,15,16,17,18]. 

There is a difference between the terms folate and folic acid. Folate is a generic term used for compounds that have similar vitamin activity to pteroylpolyglutamates, which is used to describe the vitamins found naturally in food. The term folic acid, pteroylmonoglutamic acid (PGA), represents the synthetic form added to supplements/medication and fortified foods and is used to designate the blood levels of this vitamin [19].

Regardless of the source, the importance of folate from foods and folic acids found in fortified foods is due to the fact that they can directly influence the levels of folic acid and Hcy in blood, which could be affected by polymorphisms in genes encoding enzymes related to folate metabolism and absorption (*MTHFR*, among other genes) [19]. Therefore, lower folate intake should be avoided in adult women, especially those with this polymorphism, through daily consumption of adequate amounts of this nutrient (400 µg of folate). It could be a way to maintain the immune system balanced, positively impacting the blood folic acid levels, and providing anti-inflammatory and antioxidant effects against inflammation [20,21,22].

Thus, the aim of this original study was to verify the influence of the *MTHFR* C677T polymorphism on the effect of diet intervention with folate from food sources on inflammatory biomarkers in women with overweight or obesity, providing dietary recommendations according to the genetic profile to improve health and longevity.

## 2. Materials and Methods

### 2.1. Study Characterization

A double-blind, randomized, eight-week intervention study related to the population-based study “Second Cycle of Diagnosis and Intervention on the Diet, Nutrition, and Most Prevalent Non-communicable Diseases in the Population of João Pessoa, Paraíba” (II DISANDNT/PB) was initiated in May 2015 and completed in May 2016 [23,24].

In this study, questionnaires were used to collect information about the socio-economic, demographic, and epidemiological status, anthropometric evaluation, lifestyle, 24-h dietary recall (24HR), and hematological and biochemical examinations. Thus, the questionnaires provided a rich database that encompassed all socioeconomic levels and age groups and were representative of the population of the eastern and western zones of João Pessoa city, Paraíba.

Based on information from the city, it was found that a number greater than 10,000 inhabitants and sample fractions less than 5% were needed; therefore, a finite population correction factor was unnecessary. Thus, setting the minimum sample required for an estimation of the population parameter to a level of reliability of 95% (corresponding to a critical value table Z (α/2) of 1.96) was performed using the following calculation procedure [25].
(1)$$n^{*}=\frac{{\hat{\sigma }}^{2}Z_{\alpha /2}^{2}}{E^{2}}$$

Therefore,
(2)$$n^{*}=\frac{{2.601,93}^{2}\times {1,96}^{2}}{332^{2}}\approx 236$$

### 2.2. Ethics Statement

After assessing the inclusion criteria and genotypes for the *MTHFR* gene containing the C677T polymorphism, the researchers invited individuals to participate in the intervention, explained the aims of the study which were in accordance with the ethical guidelines, and obtained signed written consent from those who agreed to participate. The study was conducted in accordance with the Declaration of Helsinki. The study was submitted to and approved by the Research Ethics Committee of the University of Paraíba under protocol number 0569/15 and was registered in Clinical Trials under the ID number: NCT03186196.

### 2.3. Population and Sampling

The intervention was carried out using a randomly selected sample based on articles published on dietary intervention versus polymorphism [26,27,28]. The final sample comprised 48 women with overweight and obesity, with ages between 20 and 59 years who were participants of the II DISANDNT/PB study. Thus, setting the minimum sample required for the estimation of the population parameter, the following calculation procedure was used:µ = ((Ζα/2 + Ζ β/2)^2^ × (dp1^2^ + dp2^2^))/(µ1 − µ2)^2^(3)

Using the statistical program R and the effect size model of Cohen [29], as well as adopting the parameters of means and standard deviation (SD), we calculated the ex-post statistical power of the sample (whose power benchmark was at least 80%). The effect size test result is shown in Table 1 and Table 2.

As shown in Table 1 above, after the ex-post effect size test, when group separation took place and the power of the sample was analyzed, all genotypes in Group 2 (191 µg/day of folate from the 300 g of vegetables and legumes) presented a power of significance higher than 80%, showing that 8 people per genotype were sufficient for the results presented in this post-intervention group.

In Group 1, the power of significance was not observed in individuals with the CC genotype; however, for individuals with the TT genotype, the power of significance was greater than 80% and approximately 80% for individuals with the CT genotype. 

### 2.4. Inclusion and Exclusion Criteria

The inclusion criteria were: women with BMI > 25 kg/m^2^ from different socioeconomic conditions and with preserved cognitive state. The exclusion criteria were set as follows: adult women alcoholics, smokers, or neuropsychiatric disorder-diagnosed patients using medications (e.g., prednisone, hydrocortisone, dexamethasone, chloramphenicol, and acetylsalicylic acid) that interfere with folic acid metabolism (during the last three months), using a multivitamin, mineral, anorexigenic, or anabolic supplement, diagnosed with chronic diseases affecting the endocrine and metabolic system, and pregnant or planning to become pregnant during the study period.

### 2.5. Experimental Protocol

Initially, the women were evaluated based on the eligibility criteria, and then, the women who consented to participate were advised to maintain a stable weight, eating habits, and levels of physical activity with respect to the data found at the baseline [30]. In addition to these guidelines, they underwent a baseline assessment with an individual dietary plan one week before beginning the dietary intervention based on the Therapeutic Lifestyle Modifications, US National Cholesterol Education Program (NCEP) [31] and American Heart Association (AHA) models [32].

Participants who wanted to change their eating habits except for folate, the frequency of physical activity or body weight during the study period were excluded from the study [30]. All the individual’s medical therapies remained unchanged throughout the study.

After application of the first R24h, biochemical tests were performed, and the C677T polymorphism in the *MTHFR* gene was analyzed; then, after 15 days, another R24h was applied to enable an analysis of the food intake before the intervention period. Thus, the participants were submitted to the baseline to standardize food consumption before the beginning of the dietary intervention.

At the end of the one-week intervention, each participant among the 48 women were randomized into intervention groups: Group 1 (G1): 24 women, subdivided in three subgroups of eight by genotypes (normal homozygous CC genotype, heterozygous CT genotype, and homozygous TT genotype); same for Group 2 (G2): 24 women, subdivided in three subgroups of eight by genotypes (CC genotype, CT genotype, TT genotype). The randomization was performed using STATA software version 14.0 for Windows.

To control the usual food intake, different criteria were respected. The total energy expenditure was calculated based on the Dietary Reference Intakes (DRIs) [21,33]. The macronutrients were distributed as recommended by the AHA [32]. The equivalent system proposed by Costa et al. [34] was used to calculate, analyze, and evaluate the intake of nutrients of the recommended diet: carbohydrates: 45%–65% (55% recommended); protein: 10%–35% (15% recommended); and total fat: 25%–35% (30% recommended).

In addition, both experimental groups received individually designed diets with nutritional guidance (containing folate-rich vegetables and grains in addition to other food groups) to control daily folate intake. The folate intake was evaluated by four 24 h recalls, analyzed by the DietWin nutritional software (Porto Alegre, RS) and the multiple source method (MSM).

Individuals from both groups were encouraged to consume folic acid-fortified foods to achieve Estimated Average Requirement, EAR (≥400 µg/day folate); Group 1 (191 µg/day folate) and Group 2 (95 µg/day folate). These quantities were based on the Diabetes and plant food products (DIAPLANT), a randomized prospective controlled trial study of Switzeny et al. [30], who analyzed the effect of a diet containing 153 μg ± 82.32 μg of folate from 300 g of vegetables in women, observed good results with an increase in methylation of the DNA promoter region in the mutL homolog 1 (MLH1) gene in individuals with diabetes mellitus type 2 (DMT2).

### 2.6. Dietary Intervention Composition

Standardization of the weighing and preparation of the vegetables used in this intervention study was carried out by a commercial restaurant in the city of João Pessoa, whose specialty is the sale of salads, supervised by a professional nutritionist. Vegetables were duly packed in individual plastic containers, transported with adequate refrigeration, and delivered daily to the participants’ residence for two months (8 weeks) by the responsible researchers, who were not aware of the food, quantities or group of each participant.

In this double-blinded study, neither the nutritionists nor participants, who were divided into two groups, had access to the following information: quantity and type of vegetables of each container, and which group received the high and low dose of folate. Only the owner of the restaurant knew all this information and labeled the containers with number 1 or 2, not necessarily indicating Groups 1 or 2.

### 2.7. Data Collection

The II DISANDNT/PB team, comprising researchers from the Graduate Program in Nutrition Sciences, University of Paraíba, were duly trained at the beginning of the data collection period. This team was responsible for the application of the food consumption questionnaires (R24h and food frequency questionnaire) and the biochemical analyses during home visits. 

For nutrition assessment, the weight and height measurements were collected in triplicate to average these three values. The body mass index (BMI) was calculated using the body-weight formula (kg) divided by the square of the height (meters) [35].

Biochemical analyses of folic acid, vitamin B12, Hcy, TNF-α, and the pro-inflammatory cytokines interleukins IL-1β and IL-6 were performed at the end of the application of the fourth R24h. An anthropometric evaluation was performed during the data collection from the II DISANDNT/PB study and after the nutritional intervention.

Hcy levels were measured using high-performance liquid chromatography (HPLC), the media plasma homocysteine for women was 7.2 μmol/L [36]. The Access Folic Acid assay is a paramagnetic particle, chemiluminescent immunoassay for the quantitative determination of folic acid levels in human serum and plasma (heparin) or red blood cells (RBC) using the Access Immunoassay Systems. Folic acid levels in serum and plasma or RBC are used to assess folate status. The serum folic acid level is an indicator of recent folate intake, while a low RBC folate value could indicate a prolonged folate deficiency. Thus, serum levels of folic acid were estimated using the commercial kit (Access Folic Acid Kit [A98032]; Beckman Coulter, Fullerton, CA, USA). Folate deficiency was defined as <3.10 ng/mL with an analytical sensitivity of 0.5 ng/mL, and the laboratory reference value for vitamin B12 is 211–911 pg/mL [37]. The quantitative dosages of TNF-α and interleukins were determined using 25 μL of the sample serum incubated for 3 h at room temperature with shaking using a chemiluminescent solid-phase IMMULITE/IMMULITE 1000 system (Siemens Medical Solutions Diagnostics, Los Angeles, CA, USA). The reference value for TNF-α was <8.1 pg/mL. For interleukins, most laboratories indicated the normal reference values as less than 5.0 pg/mL for IL-1β and 0 to 5.9 pg/mL for IL-6 [38,39,40,41,42].

### 2.8. Food Consumption

The evaluation of food consumption was carried out by applying four R24h, one weekend, and one weekday, performed with a 15-day interval before the beginning of the intervention. The aim of this procedure was to establish the participants’ dietary habits as a basis for the menu definition and intake comparison at the end of the study. A third R24h was performed after the plateau week to investigate adherence to the nutritional instructions that were given. A fourth R24h was performed at the end of the eight-week intervention to analyze the participants’ usual caloric and nutrient intake to establish whether they had adhered to the recommendations and whether changes in their global and folate dietary intake had occurred after the intervention [3].

The recalls were completed by the staff according to the reports of the interviewed individuals, including the following information: mealtime and identification of the drink or food ingested with their specific characteristics, such as the type, ingredients of the preparations, brand, method of preparation, amount consumed, portion size and home measurements.

To effectively quantify the size of the portions consumed and reduce the possible memory faults of the interviewees, an album with food figures representing the home measures at different dimensions (small, medium, large, and extra-large) that were drawn according to the real weight of the average consumption of the food validated for this population was used [43,44].

All ingredients and amounts of the food preparations were detailed, and the foods were analyzed by Dietwin (Porto Alegre, Rio Grande do Sul, Brazil), a nutrition software platform containing approximately 5230 foods and various recipes. All the registrations were based on the fourth version of the TACO and DIETWIN tables, a compilation of the following tables: Brazilian Institute of Geography and Statistics (IBGE), United States Department of Agriculture (USDA), CENEXA, German, General Repertory of Food and Revenue Technical Data Sheets. The USDA table [45] and Dietwin were used to analyze folate food consumption.

To estimate the variability of the habitual/daily intake of nutrients and to correct the intrapersonal variance, we used the R24h recalls and Multiple Source Method (MSM), a new statistical method from the European Prospective Investigation into Cancer and Nutrition [37,38,46] to estimate the usual dietary intake of nutrients and foods for populations and individuals, including episodically consumed food. Available online (https://msm.dife.de/Tps/msm/) [47].

### 2.9. Collection and Isolation of Leukocyte DNA

The leukocytes used to isolate DNA were obtained by a venous puncture to collect whole blood in sterile 4-mL tubes with 7.2 mg of K3--ethylenediaminetetraacetic acid (EDTA). All blood samples collected were analyzed at the Department of Molecular Biology of the Federal University of Paraíba (UFPB) following the protocol adapted from Miller, Dykes, Polesky [48].

Initially, to lyse red cells, lysis solution 1, containing 10 mM Tris-HCl at pH 8, 5 mM EDTA, 0.3 M sucrose, and 1% Triton-X-100, was used, followed by centrifugation at 3200 rpm and discarding the supernatant.

To obtain the precipitate of pure leukocytes, the procedure with lysis solution 1 and centrifugation was performed three times. The leucocyte precipitate was then resuspended in lysis solution 2, containing 10 mM Tris-HCl at pH 8, 0.5% sodium dodecyl sulphate (SDS), 5 mM EDTA, 0.2 μg proteinase K (Invitrogen, Carlsbad, CA, USA), and incubated in a water bath for 7 h at a temperature of 55 °C. Next, 500 μL of the aqueous solution containing 1 mM EDTA and 7.5 M ammonium acetate was added, with further centrifugation at 14,000× *g* (gravity force) for 10 min at 4 °C. Next, 700 μL of the supernatant was transferred to another sterile tube, and then, 540 μL of isopropanol was added to precipitate the DNA.

Finally, DNA was extracted from the precipitate using the following procedure: washing with 70% ethanol, centrifugation for 5 min at 12,000× *g* (gravity force), drying, and resuspension in Tris-EDTA buffer pH 8.0 [48]. If it was not possible to perform all these procedures for DNA extraction on the same day, the samples were stored in the freezer at −20 °C.

### 2.10. Analysis of the C677T Polymorphism of the MTHFR Gene

Two primers [4] were used:5′-TGAAGGAGAAGGTGTCTGCGGGA-3′ (sense);5′-AGGACGGTGCGGTGAGAGTG-3′ (antisense).

Next, the amplification process was conducted in a thermocycler using an initial denaturing condition at 94 °C for 10 min, followed by 35 cycles of denaturation at 94 °C for 30 s, annealing at 61 °C for 30 s, extension at 72 °C for 30 s, and a final extension step at 72 °C for 10 min.

Thereafter, a 198-bp product was obtained and was subsequently digested with HinfI enzyme, which recognizes and cleaves the polymorphic T allele by dividing it into two fragments: a 175 bp fragment and a 23 bp fragment. The ancestral allele C was 198 bp in size.

For the analysis of the genotypes, the samples were analyzed on 12% polyacrylamide gels. Finally, the gel was colored with 0.5% silver nitrate to allow the evaluation of the genotypes [49].

### 2.11. Statistical Analysis

The characteristics of the sample were expressed using descriptive statistics. Biomarkers were described as means and standard deviation or log-transformed and expressed in geometric mean and 95% CI.

It was verified that the distribution of the data was not strongly deviated from asymmetry. Therefore, the Central Limit Theorem was used, which states that for a sufficient sample size, n = 48 in the current study, the sample mean has an approximately normal distribution. The symmetry of the biomarkers was verified separately, one by one, adapting to the use of the Central Limit Theorem.

To analyze the initial and final values after the dietary intervention, the normal and non-normal biomarkers were analyzed according to Student’s t-Test. Student’s t-test can be used for non-normal biomarkers, provided that for the sample size, the biomarker is close to normality so that the sample mean has a normal distribution with the same mean of the biomarker and variance as the variance of the biomarker divided by the sample size. The Central Limit Theorem states that for a significantly large sample, the sample mean has an approximately normal distribution.

To compare the biomarkers of the two groups by genotyping, we used the paired student t-test, making a peer-to-peer analysis of individuals from different groups and genotypes before and after the intervention. All statistical analyses were performed using Software R version 3.3.2 (Available at https://cran.r-project.org/), adopting the significance level of 5% to reject the null hypothesis.

## 3. Results

Two hundred and thirty-eight (n = 238) women classified as overweight or obese were screened for eligibility, but 190 did not meet the inclusion and exclusion criteria, resulting in a sample of 48 women. Participants were randomly assigned in the G1 (n = 24) or G2 (n = 24) group. The final sample comprised 48 women (Figure 1).

### 3.1. General Characteristics of the Participants by Groups

The final sample was composed of 48 overweight and obese women between the ages of 20 and 59 years for both groups (G1—95 µg and G2—191 µg).

In Table 3, for all the analyzed variants (Folate, Folic Acid, Vitamin B12, Hcy, TNF-α, and IL-1β), there was no significant difference between the groups before the dietary food intervention (*p*-value > 0.05). The data showed that the groups were homogeneous at this point.

Regarding the BMI, in G,1 we observed that 54% of the sample was lower than the described mean and 45.8% had results higher or equal to the average. For G2, 33% presented lower BMI when compared to the mean, and 66.7% higher or equal to the average BMI described in the table. The lower limit was 26 kg/m^2^ and 27 kg/m^2^ and higher 38 kg/m^2^ and 50 kg/m^2^ for G1 and G2, respectively.

As for groups 1 and 2, it was observed that both of them presented reduced values of habitual folate intake according to the EAR (400 mcg/day). For both groups, 37% of the samples obtained results higher than the average and 63% had values lower than the average. Regarding the results of blood folic acid, 70% (G1) and 62% (G2) showed values according to the reference. When analyzing the results for vitamin B12, it was observed that in G1, 30% presented lower values when compared to the reference, and 58% of the samples were in agreement with the reference. For G2, 37% were lower than the reference and 71% presented results according to the reference.

We also observed that 54% of the samples in G1 reached boundary values for the marker Hcy, and for G2, 75% had higher values and 20% of the samples presented lower values when they were compared to the average. TNF-α presented 45% and 42% of the results higher than the reference for G1 and G2, respectively. For the last variable, IL-6 marker was higher in both groups, 54% in G1 and 42% in G2.

The mean for the IL-1β value in G1 pre-intervention was considered within the normal range (<5.0 pg/mL), but 50% presented results higher than the reference, while for G2, 62% had values higher than the reference. The mean of the IL-1β value in G2 pre-intervention was considered elevated (>5.0 pg/mL).

### 3.2. Effect of Folate Intervention on Different Genotypes of MTHFR C677T and Inflammatory Biomarkers

This novel intervention with folate from food sources positively interfered with the inflammatory status of the sample when the *MTHFR* genotypes were compared before and after the intervention, predominantly influencing Group 2 (191 µg/day) and the individuals with the TT genotype with a reduction in Hcy, TNF-α, IL-6, and IL-1β. It was also observed that the folate intake and folic acid serum levels were increased in all genotypes of this group. In Group 1 (95 µg/day), there was an increase in folate intake in individuals with the CC and TT genotypes with higher blood folic acid concentrations for individuals with the CT and TT genotypes. Although there was no reduction in Hcy or inflammatory parameters in TT or other genotypes. The best results in inflammatory parameters and Hcy values were with the 191 μg/day diet in the *MTHFR* 677C > T dependent genotypes, because of a possible increase in circulating folic acid in the blood.

Table 4 shows that higher concentrations of folate intake had a significant impact on the response of folic acid in the blood, homocysteine concentrations, and inflammatory parameters.

When analyzing the results from the biomarkers, G1 (95 µg/day folate) showed high levels of Hcy in individuals with the TT genotype, with a small and non-significant reduction (*p* > 0.005), and IL-1β levels that were borderline for individuals with the CT genotype, presenting reduction. In G2 (191 µg/day of folate), individuals with the TT genotype had a significant reduction in Hcy concentration (*p* < 0.0005), stabilizing at borderline mean levels. Regarding IL-6 and IL-1β, the levels were borderline and reduced in individuals with the TT genotype. These results already show signs of subclinical inflammation.

In the diet with 95 μg/day (Group 1), a significant increase of folate occurred in individuals with the CC and TT genotypes. For folic acid, a significant elevation was observed in individuals with the CT genotype, and a slight increase in individuals with the TT genotype. However, the impact of such elevation on inflammatory parameters could only be observed discreetly in individuals with the CT genotype.

On the other hand, in the diet with 191 μg/day (Group 2), the elevation of folate intake and folic acid metabolites was significant in all genotypes. However, the reduction in homocysteine levels and all the inflammatory markers’ activity occurred only in individuals with the TT genotype, suggesting that the benefits from the diet, at least as far as inflammation is concerned, are genotype-dependent.

Regarding the results of vitamin B12 in both groups, we did not observe any change in this parameter when comparing pre- and post-dietary intervention.

## 4. Discussion

Many studies have shown that an intervention with enriched folic acid diet resulted in positive health effects in adults with hyperhomocysteinemia and low folic acid levels carrying the *MTHFR* C677T polymorphism [50,51,52,53,54,55,56,57]. However, to date, no other study has presented significant results from a folate intervention on inflammatory biomarker levels of women carrying the *MTHFR* C677T polymorphism.

The folate diet was obtained from vegetables and legumes, offering an estimated average requirement (EAR) between 87.5% and 102.5% (95–191 μg/day) over eight weeks. This diet was able to reduce Hcy, inflammatory biomarker levels, and also increase the folic acid concentration in young women with overweight or obesity. The effectiveness and adhesion to the intervention were facilitated by the researchers that provided an accessible intake of the five vegetable and legume servings/day (300 g per day) as recommended by the “Therapeutic Lifestyle Changes” model, US National Cholesterol Education Program (NCEP) [31], and American Heart Association [33].

Regarding folic acid content, individuals with the 677TT genotype from both groups (1 and 2) presented increased levels, but with different intensities—higher in G2 with an intake of 191 µg/day (*p* < 0.005) than in G1 containing 95 µg/day (*p* < 0.05)—likely as a result of the different amounts administered in the intervention. This result corroborates those obtained by Tsang et al. [6] with a meta-analysis of trials and observational studies using folic acid supplementation. 

Similar outcomes were found by Brouwer et al. [58] in a randomized control trial with 144 healthy women in Holland. The long-term folic acid supplementation (250 and 500 μg of folic acid/day) for four weeks was able to increase the concentration of folic acid in serum and red blood cells (RBC). In another study conducted by Arias et al. [59], with a group of 34 young Colombian women at reproductive age, the supplementation of 400 μg/day of folic acid for three months showed a better response regarding folate intake and folic acid blood levels in individuals with the 677TT genotype when compared to the 677CC or 677CT genotypes. Corroborating these results, Anderson et al. [60] demonstrated with 142 participants that the increase in folic acid concentrations with 400 µg supplementation depended on the *MTHFR* C677T genotype, with the 677TT genotype presenting a better response than individuals with the 677CC or 677CT genotypes.

In the present study, 54% of the G1 and 75% of the G2 reached the borderline reference values for Hcy, however, 20% of the G2 presented values below the borderline reference. Interestingly, Hcy levels were decreased only in G2 in individuals with the TT genotype, indicating that for this purpose, the diet was only effective under conditions where the folate bioavailability was low, i.e., in carriers of two alleles encoding the enzyme in the thermolabile version [61]. Hcy is a metabolite of the methionine remethylation pathway and has been considered a potential marker of methyl-THF formation, acting as a methyl group donor. Studies have indicated that higher Hcy levels (approximately 20%) are associated with low folate levels in individuals with the *MTHFR* TT genotype. In this context, the *MTHFR* TT has a reduction of 70% of its catalytic activity caused by the C677T polymorphism [4]. Similar results of homocysteine concentrations demonstrating reduction after long-term supplementation with folic acid were found in many other studies [58,59,62,63], especially in individuals with the 677TT genotype [55,60,64].

We also observed in our sample that obesity/overweight (elevated BMI and WC), associated with hyperhomocysteinemia caused by C677T polymorphism, can trigger inflammation and oxidative stress by activating the NF-κB signaling pathway [65,66]. Furthermore, the production of Reactive Oxygen Species (ROS) could selectively change the pattern of interleukins’ expression [67,68,69] and tumor necrosis factor-alpha (TNF-α). TNF-α is an important inflammatory marker that induces the synthesis of cytokine mediators IL-1β and IL-6 [70,71], acting synergistically and intensifying inflammation [72,73,74]. Therefore, to disrupt the vicious cycle that could lead to the development of chronic disease, the reduction of this inflammation in women with the C677T polymorphism in the *MTHFR* gene is very important [75,76,77]. In the present study, we can infer that the folic acid intake did not alter the weight of women.

It is not surprising that the most robust response in terms of reduction of inflammatory mediators occurred under the conditions in which Hcy was decreased significantly—i.e., in individuals with the TT genotype and diet of 191 μg/day. It was suggested that the observed effect occurred because, in individuals with the 677TT genotype only, there is a folate deficit. The deficit is caused by the *MTHFR* with reduced activity in the remethylation pathway and elevated Hcy, consequently leading to CH3-folate-limiting levels. Thus, in the present study, folate availability was sufficient to significantly reduce Hcy using supplementation longer than four weeks. The same results were demonstrated in the meta-analysis carried out by Colson et al. [64] and confirmed by other studies [3,15,55,56,60,64,76], showing a reduction in inflammatory parameters, and suggesting that one effect is a consequence of the other.

In a previous work of the researcher’s group presented by Ribeiro et al. [3], a similar intervention with vegetables and food was performed, however, other nutrients such as vitamin B12 (cofactor in One Carbon Metabolism), vitamin A and C, selenium, etc. were evaluated. As a result, they did not observe a significant difference between the genotypes and these other nutrients. Other related nutrients like choline were not analyzed because that was not the purpose of the study.

For all these reasons, we suggest an antioxidant-rich diet for young women with *MTHFR* C677T polymorphism. It provides essential nutrients like folate, an important vitamin in the regulation of inflammatory response, cell damage, DNA repair, antioxidant capacity (CAT), Hcy levels, and stability of the MTHFR enzyme [3,16,55,78,79,80]. Considering that nutrigenetics explains the interaction of gene-nutrients [81,82], our results corroborate with several studies that have reported the importance of personalized nutritional intervention reducing C677T polymorphism effects and preventing many diseases [15,17,27,83,84,85].

Specifically in our study, participants tolerated well the amounts of folate-rich foods and vegetables offered. Based on many studies previously cited in the present work, we observed the use of folic acid supplementation, folate fortification, vegetable consumption, etc.; however, we did not see many studies with low folate doses (<400 mcg/day) with good results. Effective changes in dietary habits that incorporate adequate amounts of nutrients such as folate need to involve educational processes, requiring investments and government efforts. According to the literature consulted, the intake of folate does not supply the EAR recommendation, so in countries where folic acid fortification is not available, insufficient intake of folic acid is more pronounced [19]. Thus, greater inadequate folate consumption would contribute to more detrimental health outcomes in individuals with the *MTHFR* C677T polymorphism.

According to the “Sakado Folate Project” conducted by Kagawa et al. [57], the genotype-based personalized diet was better accepted than the general individual diet. The personalized diet was more efficient to motivate change in the participants’ lifestyle and improve their nutritional status by elevating green leafy vegetables and folic acid-fortified food intake even one year after the end of the study. Nielsen; El-Sohemy [86] and Hiraoka; Kagawa [17], observed a similar effect with this type of intervention connecting genotype and folate content.

Previous studies cited above have demonstrated a high association between hyperhomocysteinemia, folic acid blood levels, and development of oxidative stress, especially for individuals with the TT genotype. However, this was the first study to show not only elevated folic acid concentrations and decreased Hcy levels but reduced inflammatory biomarkers in individuals with the C677T allele after intervention with folate.

The limitation of the study was the sample size. However, we still observed a dietary response with the highest dose of folate, even under the influence of the *MTHFR* C677T polymorphism. Thus, the results of this study can demonstrate the importance of considering individual information when establishing nutritional goals for specific biological responses. The control group was not used in this study because the researchers wanted to investigate which dose of folate in the natural diet had benefits for obese adult women with C677T polymorphism.

## 5. Conclusions

Based on our original results, the intervention with folate consumption (191 µg/day) from natural foods for eight weeks had a beneficial effect on the reduction of inflammatory markers (TNF-α, IL-6, and IL1β) and Hcy in overweight and obese women with *MTHFR* C677T polymorphism TT genotype. It is important to highlight why biological individuality should be considered a priority to improve people’s health.

## Figures and Tables

**Figure 1 nutrients-12-00361-f001:**
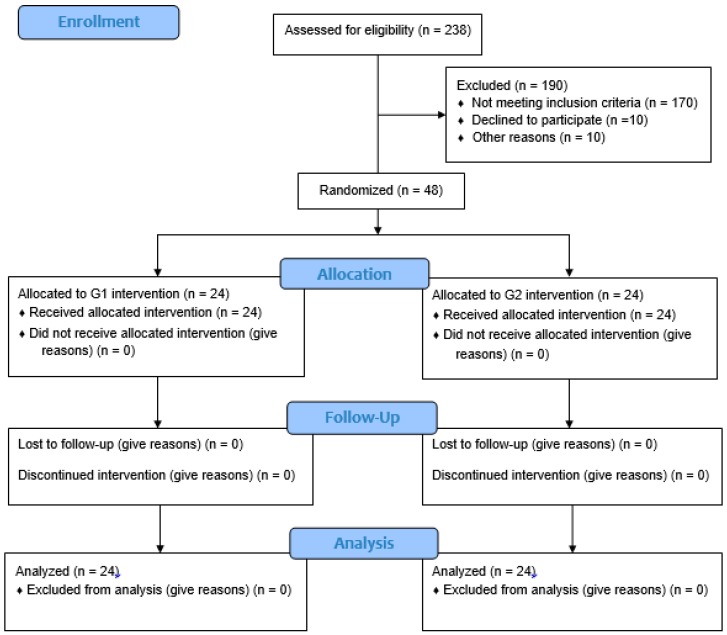
Participant flow through the study. G1, Group 1; G2, Group 2.

**Table 1 nutrients-12-00361-t001:** Size Effect.

Group	Gene	est_d	n	Significance	Power
2	CC	1.48	8	5%	94.5%
2	CT	1.96	8	5%	99.7%
2	TT	1.89	8	5%	99.5%
1	CC	0.43	8	5%	18.5%
1	CT	0.69	8	5%	78.7%
1	TT	1.40	8	5%	92.1%

**Table 2 nutrients-12-00361-t002:** Analysis of the mean and standard deviation.

Group	Gene	n	SD_Before FA	Mean_Before FA	SD_After FA	Mean_After FA
2	CC	8	5.30	12.95	7.27	22.22
2	CT	8	1.62	13.49	3.10	18.11
2	TT	8	5.12	16.53	4.36	25.48
1	CC	8	4.09	13.18	3.79	14.89
1	CT	8	3.98	13.32	4.96	16.40
1	TT	8	4.60	15.01	6.30	22.63

FA = Blood Folic Acid; SD = standard deviation.

**Table 3 nutrients-12-00361-t003:** General characteristics of the participants by groups before the feeding intervention.

Biomarkers	Group 1 (95 µg/Day)	Group 2 (191 µg/Day)	*p*-Value
Age (years)	44.33 ± 9.26	44.88 ± 12.47	0.8802
BMI (kg/m^2^)	30.47 ± 4.82	29.88 ± 3.01	0.6236
WC (cm)	91.78 ± 12.02	91.63 ± 7.96	0.9600
Folate (µg)	150.90 ± 42.73	144.40 ± 43.47	0.6034
Folic Acid (μg/mL)	13.83 ± 4.13	14.32 ± 4.46	0.6222
Vitamin B12 (pg/mL)	304.66 ± 133.05	253.14 ± 103.58	0.1417
Homocysteine (μmol/L)	11.65 ± 5.99	9.64 ± 2.44	0.1404
TNF-*α* (pg/mL)	6.14 ± 7.26	6.51 ± 7.90	0.8661
IL-1β (pg/mL)	3.88 ± 3.90	5.11 ± 4.92	0.3429
IL-6 (pg/mL)	4.42 ± 3.85	4.49 ± 3.47	0.9894

Values expressed as mean and standard deviation (±SD). TNF-α = tumor necrosis factor-alpha; BMI = Body Mass Index; IL = Interleukin; WC = waist circumference.

**Table 4 nutrients-12-00361-t004:** Values of vitamins, homocysteine, and inflammatory markers, separated by the Methylenetetrahydrofolate reductase *MTHFR* genotypes, groups and pre and post-diet intervention with folate.

**Group 1 95 µg/Day**									
	**Pre**	**Post**	***p*-value**	**Pre**	**Post**	***p*-value**	**Pre**	**Post**	***p*-value**
	**CC**	**CC**		**CT**	**CT**		**TT**	**TT**	
Folate (μg)	149.0 ± 14.43	292.0 ± 65.24	0.0004 *	170.0 ± 50.43	254.0 ± 85.27	0.0910	154.0 ± 53.84	316.0 ± 63.06	0.0003 *
FA (μg/mL)	13.18 ± 4.08	14.88 ± 3.79	0.1318	13.32 ± 3.97	16.4 ± 4.95	0.0011 *	15.00 ± 4.60	22.62 ± 6.30	0.0249 *
Vit. B12 (μg/mL)	268.6± 92.63	271.0 ± 92.16	0.3185	345.5 ± 179.14	355.2 ± 202.59	0.7991	299.8 ± 118.65	273.6 ± 83.60	0.3162
Hcy (μmol/L)	13.61 ± 13.10	7.83 ± 2.51	0.2408	10.98 ± 6.11	9.86 ± 4.43	0.6812	19.33 ± 12.75	14.91 ± 10.40	0.4603
TNF-α (pg/mL)	5.80 ± 8.33	4.45 ± 3.54	0.6489	9.52 ± 8.63	2.84 ± 2.09	0.0517	3.10 ± 2.34	4.30 ± 6.99	0.5959
IL-6 (pg/mL)	4.26 ± 5.10	4.48 ± 2.29	0.9184	5.16 ± 1.60	2.82 ± 2.96	0.0685	3.85 ± 4.39	3.29 ± 3.07	0.8310
IL-1β (pg/mL)	3.50 ± 4.44	3.85 ± 1.78	0.8280	5.43 ± 4.50	1.55 ± 1.79	0.0224*	2.71 ± 2.34	3.69 ± 4.58	0.6713
**Group 2 191 µg/Day**									
	**Pre**	**Post**	***p*-value**	**Pre**	**Post**	***p*-value**	**Pre**	**Post**	***p*-value**
	**CC**	**CC**		**CT**	**CT**		**TT**	**TT**	
Folate (μg)	132.0 ± 32.73	352.0 ± 73.07	0.0000 *	128.0 ± 41.17	410.0 ± 372.35	0.0000 *	184.0 ± 34.54	376.0 ± 68.32	0.0000 *
FA (μg/mL)	12.94 ± 5.29	22.21 ± 7.26	0.0033 *	13.48 ± 1.61	18.11 ± 3.10	0.0016 *	16.52 ± 5.11	25.48 ± 4.35	0.0046 *
Vi. B12 (μg/mL)	251.8 ± 113.13	237.5 ± 52.69	0.7174	261.2 ± 139.79	258.0 ± 113.03	0.8429	246.3 ± 52.71	228.6 ± 78.15	0.6492
Hcy (μmol/L)	8.97 ± 1.52	7.60 ± 1.79	0.1207	9.91 ± 3.70	8.01 ± 2.27	0.2374	10.05 ± 1.66	7.11 ± 0.77	0.0005 *
TNF-α (pg/mL)	7.73 ± 12.34	5.46 ± 9.38	0.5132	5.02 ± 4.59	1.66 ± 2.31	0.1614	6.78 ± 5.24	1.66 ± 0.53	0.0004 *
IL-6 (pg/mL)	3.93 ± 4.70	3.63 ± 3.41	0.8980	3.86 ± 3.76	1.84 ± 2.03	0.2224	5.67 ± 0.93	0.66 ± 0.74	0.0000 *
IL-1β (pg/mL)	3.17 ± 5.74	17.62 ± 1.63	0.2610	6.85 ± 5.82	3.01 ± 3.43	0.1565	5.32 ± 2.20	1.73 ± 2.47	0.0007 *

* All values are expressed as mean and standard deviation (±SD). *p*-values refer to the Student’s t-test for paired sample mean differences. In all t statistics, the degree of freedom of the model was equal to 15. G1: 191 µg/day of folate; G2: 95 µg/day of folate. FA: folic acid (μg/mL); Vit. = Vitamin; IL- interleukin; TNF: tumor necrosis factor; Hcy: homocysteine; CC, cytosine-cytosine; CT, cytosine-thymine; TT, thymine-thymine.

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
