# Peer review of "Food Intervention with Folate Reduces TNF-α and Interleukin Levels in Overweight and Obese Women with the MTHFR C677T Polymorphism: A Randomized Trial"

_nutrients, 2020, doi:10.3390/nu12020361_

Round 1
Reviewer 1 Report
The manuscript examined the intervention of dietary folate on pro-inflammatory marker levels of overweight and obese women with the MTHFR C677T polymorphism. This is an interesting topic that alerts nutritionists and health care providers to the importance of nutrigenomics, for individualized nutrition interventions. That said, this manuscript requires modifications and clarification of several points. Please see below:
Check the whole manuscript for the use of English, focusing especially on the Abstract and Introduction
Please distinguish between folic acid and folate, describe their importance related to your study and the rationale of measuring both.
Materials and Methods
Please give rationale for the decision of the amount of food folate included in the two diet groups.
No control group has been used and no reason has been given for its lack as part of limitations.
Since the RDA for folate is 400micrograms/day, what are the authors trying to achieve? Are the amounts given in form of veges to the experimental groups in addition to 400micrograms or attempting to reach 400micrograms?
This is not clear.
Lines 178-181 are not clear. Did the authors control the rest of the day's food intake? Did they give participants lists of foods to avoid or emphasize to reach 400micrograms?
Please be clear as to the time-line and frequency of sampling (diets, anthropometrics, blood draws). Did the nutritionists meet with the participants to review food records?
Results
A table with dietary intake of vitamins and minerals involved in the FA pathway as well as macronutrients during the different sampling times is advisable.
Please comment and present Table 3. Comment on initial levels of the biomarkers under study. Are they normal, borderline etc.? Please check with normal levels of IL-beta.
On 3.2: it seems that there is an error in what is presented on Table 4 which does not agree with your statement on lines 317-320.
The initial concentrations of the pro-inflammatory biomarkers in G1 with the CT genotype are markedly higher than the subjects with the CC and TT genotype. Has a statistical analysis (Repeated Measures ANOVA) been conducted across time points and different diet group?
Similarly with the G2 group, except for TNF-alpha, the concentration of the biomarkers are also markedly higher in the TT genotype. You may want to comment on this in the discussion.
Discussion
Even though the title refers to overweight and obese women, I do not see it discussed at all in the discussion or introduction sections in relationship to your measurements an topic explored. How are they all related?
There is need for relating your findings to previous literature and integrating in a flowing document. Please focus on your findings.
Author Response
Review 1
The manuscript examined the intervention of dietary folate on pro-inflammatory marker levels of overweight and obese women with the MTHFR C677T polymorphism. This is an interesting topic that alerts nutritionists and health care providers to the importance of nutrigenomics, for individualized nutrition interventions. That said, this manuscript requires modifications and clarification of several points. Please see below:
Check the whole manuscript for the use of English, focusing especially on the Abstract and Introduction – in process AJE.
Please distinguish between folic acid and folate, describe their importance related to your study and the rationale of measuring both.The reason for measuring both (dietary and blood folate) is that this is a fundamentally important nutrient in 1 carbon metabolism for methyl group donation that enables the remediation of Hcy in methionine. Blood values being influenced by consumption.
Thus, with the present study we would like to show that individuals with the M6FR gene C677T polymorphism, especially the TT genotype could be responsive to folate from food sources.
Thus, habitual consumption of folate intake was reported in Table 4 as well as blood folate concentrations to better elucidate this effect.
Materials and Methods
Please give rationale for the decision of the amount of food folate included in the two diet groups. These amounts of folate-rich foods were used based on the study by Switzeny et al. (2012), who analyzed the effect of a diet with 153 μg ± 82.32 μg of folate contained in 300 g of vegetables in women, observing good results from an increase in methylation of the DNA promoter region in the MLH1 gene in individuals with DMT2.
No control group has been used and no reason has been given for its lack as part of limitations. The present study aimed to analyze different but affordable amounts of dietary folate that could have an important effect on this female population. As for the absence of the control group, this was due to the difficulty in selecting another group of women eligible for the study.
Since the RDA for folate is 400micrograms/day, what are the authors trying to achieve? Are the amounts given in form of veges to the experimental groups in addition to 400micrograms or attempting to reach 400micrograms? To control the usual food intake, different criteria were respected. Total energy expenditure was calculated based on the Dietary Reference Intakes (DRIs) [20]. Macronutrients were distributed as recommended by the AHA [32]. The equivalent system proposed by Costa et al. [33] was used to calculate, analyze and evaluate the intake of nutrients contained in the recommended diet: carbohydrates: 45-65% (55% recommended); protein: 10-35% (15% recommended); and total fat: 25-35% (30% recommended). In addition, both experimental groups received individually designed diets with nutritionist guidance (containing folate-rich vegetables and grains in addition to other food groups) to control daily folate intake. This folate intake was evaluated by four 24-hour recalls, analyzed by DietWin nutritional software (Porto Alegre, RS) and the multiple source method (MSM) [36,37,45].Individuals from both groups were encouraged to consume folic acid-fortified foods to achieve EAR (estimated average requirement) (≥ 400 µg / day folate), but one (191 µg / day folate) received more than other (95 µg / day folate) folate).
This is not clear.
Lines 178-181 are not clear. Did the authors control the rest of the day's food intake? Did they give participants lists of foods to avoid or emphasize to reach 400micrograms?
Because it is the supply of nutrients via vegetables / food, we present in our study that vitamin B12, a vitamin that acts as a factor in the metabolism of folic acid in the Hcy remetylation pathway, was not statistically significant.
In addition, a more detailed analysis of the main anti-inflammatory and antioxidant nutrients in the foods provided was performed and previously published by Ribeiro et al. (2018), however, the other nutrients did not present statistical relevance that interfered with the study results.
Results
A table with dietary intake of vitamins and minerals involved in the FA pathway as well as macronutrients during the different sampling times is advisable. Yes, We analyzed other nutrients.
Fonte: Ribeiro et al. (2018)
Please comment and present Table 3. Comment on initial levels of the biomarkers under study. Are they normal, borderline etc.? Please check with normal levels of IL-beta.
In Table 3, for all analyzed variants (Folate, Folic Acid, Vitamin B12, Hcy, TNF-α and IL-1β) there was no significant difference between the groups before the food intervention (p-value> 0.05). . Showing that from this point the groups were homogeneous.
However, when stratified by genotypes and compared pre and post intervention, differences were observed.
The mean pre-intervention IL-1β value in G1 was considered within normal range (<5.0 pg / mL) and elevated for G2.
On 3.2: it seems that there is an error in what is presented on Table 4 which does not agree with your statement on lines 317-320.
In fact, there was a mistake in describing the data in Table 4. G1 (95 µg / day) showed increased folate levels in CC and TT genotypes and increased folate blood concentration in CT and TT genotypes.
Corrected paragraph:
In Group 1 (95 µg / day) there was an increase in folate intake in CC and TT genotypes with higher blood folic acid concentrations for CT and TT genotypes.
Although there was no reduction in Hcy or inflammatory parameters in TT or other genotypes. Evidence that the best results in inflammatory parameters and Hcy values were with the 191 μg / day diet in the MTHFR 677C> T dependent genotypes, due to a possible increase in circulating folate.
The initial concentrations of the pro-inflammatory biomarkers in G1 with the CT genotype are markedly higher than the subjects with the CC and TT genotype. Has a statistical analysis (Repeated Measures ANOVA) been conducted across time points and different diet group? For the construction of the statistics in table 4 we used the paired student t-test, making a peer-to-peer analysis of individuals from different groups and genotypes before and after the intervention. Although on average the values of the initial concentrations of proinflammatory biomarkers in G1 with the CT genotype are higher than individuals with the other genotypes, these initial values are within the confidence interval. Thus, we cannot reject that a priori that with 5% significance (considering the mean and its dispersion) that these individuals in distinct genotype groups have different starting points from the biomarkers.
Similarly with the G2 group, except for TNF-alpha, the concentration of the biomarkers are also markedly higher in the TT genotype. You may want to comment on this in the discussion. The case cited in Group 2 is similar to the one mentioned above, on average individuals of the TT genotype have generally higher concentrations, but they are statistically equal to 95% confidence.
Discussion
Even though the title refers to overweight and obese women, I do not see it discussed at all in the discussion or introduction sections in relationship to your measurements an topic explored. How are they all related?
There is need for relating your findings to previous literature and integrating in a flowing document. Please focus on your findings.
Obese women tend to consume foods rich in sugars, trans fats, saturated and low in anti-inflammatory / antioxidants, which generates accumulation of abdominal fat (high BMI and circumference).
In turn, this accumulated fat, according to several articles already mention, there is activation of inflammatory genes by activation of the NkkappaB pathway, producing several cytokines. Thus, with Hcy elevation, due to the presence of the C677T polymorphism, this excess amino acid in the bloodstream oxidizes generating reactive oxygen species (EROS) and inflammatory cytokines.
Thus, the association of C677T polymorphism with HCY elevation and the accumulation of excessive body fat (overweight and obesity) leads to higher production of inflammatory cytokines (such as interleukins, TNF alpha), generating a subclinical inflammation that can be seen in biochemical tests and later trigger non-communicable chronic diseases.
Thus requiring proper intervention with balanced diet especially with antioxidant and anti-inflammatory nutrients.

Reviewer 2 Report
The aim of this study was to understand the impact of supplementation on the immune response in women with the MTHFR polymorphism. I think that the authors did a good job of outlining the methods used, there are some areas of the paper that I think need some revision.
Introduction, line 63 spelling of methylenetransferase and line 66 spelling of methylation need to be corrected.
The study focuses on high levels of homocysteine causing negative health outcomes, can the authors list what these levels of homocysteine are? This should also be stated in the discussion where applicable.
For Table 4, what are the p-values from? ANOVA or paired comparison?
The authors should list the F-values and df for all results, as well as t statistic. I think this would be helpful for the reader to understand what statistics were completed.
Would other components of the one-carbon metabolism be more effective at reducing Hcy levels, such as choline or vitamin B12? This would be an interesting point.
The authors should also discuss folic acid fortification that is currently in place in several countries world wide. Would the results be more robust in countries that do not have folic acid fortification?
Overall, I think this is a well written study, but requires a few revisions.
Author Response
Review 2
The aim of this study was to understand the impact of supplementation on the immune response in women with the MTHFR polymorphism. I think that the authors did a good job of outlining the methods used, there are some areas of the paper that I think need some revision.
Introduction, line 63 spelling of methylenetransferase and line 66 spelling of methylation need to be corrected. Ok.
The study focuses on high levels of homocysteine causing negative health outcomes, can the authors list what these levels of homocysteine are? This should also be stated in the discussion where applicable. Included in the 2.7 data collection: Hcy levels were measured using high performance liquid chromatography (HPLC), the reference plasma homocysteine average for women was 7.2 μmol / L [35.], in the present study, the The mean standard deviation of HCy was 11.65 μmol / L for G1 and 9.64 μmol / L for G2. In both groups more than 80% of women had higher values.
For Table 4, what are the p-values from? ANOVA or paired comparison?
The authors should list the F-values and df for all results, as well as t statistic. I think this would be helpful for the reader to understand what statistics were completed. In table 4, the p values refer to the t-test, whose degree of freedom (df = degrees of freedom) was 15 for all estimated t statistics. For a better understanding of the statistics used, we made clear the note at the bottom of table 4. Considering that the p-value statistic expresses the hypothesis test results for mean differences that we developed in the work, we found it more efficient to display only the p-value. In addition, the inclusion of the t-statistic in each of the tests shown in table 4 would significantly increase the number of columns and make it more difficult for the reader to understand the results.
All values are expressed as mean and standard deviation (± SD). P-values refer to Student's t-test for paired sample mean differences. In all t statistics, the degree of freedom of the model was equal to 15. G1: 191 µg / day of folate; G2: 95 µg / day of folate. FA: folic acid (μg / mL); Vit. = Vitamin; IL- interleukin; TNF: Tumor Necrosis Factor; Hcy: homocysteine.
Would other components of the one-carbon metabolism be more effective at reducing Hcy levels, such as choline or vitamin B12? This would be an interesting point. Certainly it might be important to investigate other intervention modalities and homocysteine reduction. However, the focus of this study is the effect of folic acid supplementation since the analyzes showed a deficiency in consumption in the study population and the absence of studies supporting the effect on inflammation markers by verifying homocysteine response MTHFR C677T genotype dependent.
The authors should also discuss folic acid fortification that is currently in place in several countries world wide. Would the results be more robust in countries that do not have folic acid fortification?
Based on the many studies we cited, we observed the use of folic acid supplementation, folate fortification, vegetable consumption, etc., we do not see many studies with low folate dose / amounts (based on EAR 400 mcg / day), believing that it is likely that It could be more difficult to obtain the benefits offered by this nutrient with low folate consumption, ie without fortification based on our results.
Overall, I think this is a well written study, but requires a few revisions.

Reviewer 3 Report
In this study authors have evaluated the effects of a dietary intervention, and not folate supplements, on the levels of tumour necrosis factor-α (TNF-α), interleukin-6 (IL-6) and interleukin-1β (IL-1β), markers of inflammatory process, in overweight and obese women with the MTHFR C677T polymorphism.
They claim original results showing after folate consumption (191 μg / day) from a 8 week intervention with natural foods a reduction of inflammatory markers in overweight and obese women with MTHFR C677T polymorphism TT 424 genotype.
The study seems well conducted and actually, their observations seem to indicate, confirming other studies, that when folate availability was sufficient to significantly reduce Hcy, a reduction in inflammatory parameters was observed.
Therefore the suggestion of an antioxidant-rich diet to young women with C677T MTHFR polymorphism seems obvious and appropriate. In particular, this diet should include folate that is involved in the regulation of inflammatory response, cell damage, DNA repair, antioxidant capacity (CAT), Hcy levels and stability of MTHFR enzyme.
However, as the authors themselves statethe sample size is the main limitation of this study.
Nevertheless, it would demonstrate the importance of considering personalized nutrition for specific biological responses and to improve people’s health.
They emphasize the importance that this was the first study to show not even elevated folate concentration and decreased Hcy levels, but also reduced inflammatory biomarkers in 677T allele carriers after a natural folate diet intervention.
Author Response
Thanks for the comments. The language was improved.
Round 2
Reviewer 1 Report
Thank you for addressing the concerns of this reviewer but many of the authors' responses do not directly answer the question.
The authors have not incorporated the suggestions of this reviewer for strengthening the discussion and/or intro sections.
The authors have not added limitations to this study, such as the lack of control.
Author Response
Dear reviewers,
We kindly inform that the English review was made by American Journal Experts and we are sending the certificate attached. However, we imagine that a new revision of English is being requested for a native translation. Then, in this case if you consider it really necessary we’ll accept the suggestion and we could submit the paper’s final version to the MDPI's English editing service.
Below are updates on the introduction, discussion, and absence of control group sessions.
Introduction, paragraph 1 (lines 58-63) the overweight/obesity relation: Women with an unbalanced diet in calories can have excessive accumulation of body fat specially in the abdominal circumference, elevated body mass index (BMI) and weight gain. This overweight and obese status, hides a subclinical inflammation with important production of cytokines by the activation of NF-κB pathway. Also, in the presence of increased levels of homocysteine (Hcy) caused by the MTHFR gene single nucleotide polymorphism (SNP) C677T, the inflammation and oxidative stress can be potentiated, changing the metabolism homeostasis [1-3].
Introduction, paragraph 1 (lines 72-76) the overweight/obesity relation Furthermore, high Hcy oxidation and body fat accumulation could lead to cytokine-mediated inflammation and increase of reactive oxygen species (ROS) production and oxidative stress for activating the NF-κB signaling pathway, requiring a balanced diet rich in antioxidant and anti-inflammatory nutrients, like folate that could act reducing this damage and also possibly would neutralize the C677T polymorphism effects [3,13-18].
Justifying: We have strengthened the relationship of overweight / obesity with regard to some damage that the MTHFR gene C677T polymorphism can cause along with low folate consumption, directly reflecting in the reduction of folic acid values in the blood.
We then explain the action of this polymorphism, the activation pathway of inflammation as the possible damage to the organism of these individuals, the occurrence of this polymorphism due to low folate consumption along with inflammation and oxidative stress caused by excess body fat, would generate more damage to health.
Thus, a diet with a possible anti-inflammatory nutrient (folate) in the appropriate amounts as recommended would contribute to reducing or neutralizing these damages, providing more health and quality of life for these people through individualized nutrition.
We also added the folate and folic acid distinction, as well as explaining our intent to measure them in this study. Please find it below:
Introduction (lines 77-89): There is a distinction between dietary folate and folic acid, folate being the generic term used for compounds that have similar vitamin activity to pteroy-L-glutamic acid which is used to describe the forms of this naturally occurring vitamin in food and the term folic acid represents synthetic form found in medicated supplements, fortified foods and to designate this vitamin in the blood.
In the present study, the importance of the consumption of dietary folate and folic acid found in fortified foods is due to the fact that this consumption directly influences the concentrations of folic acid and Hcy in blood, which in turn can be affected by polymorphisms in genes encoding enzymes related to folate metabolism and absorption (MTHFR among other genes) (Shi, Caprau et al., 2003).
Results section (lines 330- 350), for Table 3: We also described better the Table 3, regarding all the variants analyzed (age, BMI, waist circumference (WC), Folate, Folic Acid, Vitamin B12, Hcy, TNF-α and IL-1β).
In Table 3, for all analyzed variants (Folate, Folic Acid, Vitamin B12, Hcy, TNF-α and IL-1β) there was no significant difference between the groups before the food intervention (p-value> 0.05). Showing that from this point the groups were homogeneous.
Regarding BMI, 54% and 33% of the sample in G1 and G2 had lower BMI than the described mean; 45.8% and 66.7% had a BMI greater than or equal to the average BMI described in the table, with the lower limits being 26 kg / m² and 27 kg / m² and higher than 38 kg / m² and 50 kg / m² for G1 and G2, respectively.
As for groups 1 and 2, it was observed that all presented reduced values of habitual folate food intake according to the EAR (400 mcg / day); In both groups, 37% had values above the average found in the present study and 63% had values below the average. Regarding the results of blood folic acid 70% and 62% presented results according to the reference values; 30% and 37% below the reference values; for vitamin B12, 58% and 71% presented results according to reference values, for G1 and G2, respectively.
The marker Hcy presented 54% of reference boundary values for G1; for G2, 75% had values above the average and 20% below. In relation to TNF-α, 45% for G1 and 42% for G2 presented results higher than reference values. For the IL-6 marker, 54% and 42% presented high values for G1 and G2, respectively.
The mean pre-intervention IL-1β value in G1 was considered within the normal range (<5.0 pg / mL), but 50% presented results higher than reference values; for G2 62% presented results higher than reference values. The mean pre-intervention IL-1β value in G2 was considered elevated (>5.0 pg / mL).
Discussion: The relation with the overweight/obesity could be found, respectively in lines 428-437.
We also observed that the presence of obesity/overweight (elevated BMI and WC) in our population, associated with hyperhomocysteinaemia caused by C677T polymorphism, can trigger oxidative stress and inflammation by activating the NF-κB signaling pathway [66-68] and producing reactive Oxygen Species (ROS) that can selectively change the expression pattern of interleukins [68,69] and tumor necrosis factor alpha (TNF-α). TNF-α is an important inflammatory marker that induces the cytokine mediators IL-1β and IL-6 [71,72] acting synergistically and potentializing inflammation [73,74,75]. Therefore, it is important to reduce this inflammation in women with C677T polymorphism in the MTHFR gene to disrupt the vicious cycle that can lead to chronic disease [76,77]. In the present study, we can infer that there was no statistic effect of folic acid consumption on the weight of the sample.
Discussion (lines 417-418) about Hcy: The marker Hcy presented 54% of reference boundary values for G1; for G2, 75% had values above the average and 20% below.
Discussion (lines 447-451): other nutrients analyzed.
In the case of an intervention with nutrients from vegetables and food, other nutrients such as vitamin B12 (cofactor in 1 carbon metabolism), vitamin A and C, selenium, etc., were also evaluated in a previously work of the researchers group presented by Ribeiro et al. [3]. As a result, there was not observed a significant statistic difference between the genotypes and the intake. Others related nutrients like choline was not analyzed, because it was not the purpose of the study.
Discussion (lines 459-468) about fortification, please found our explanation or below:
Specifically in our study, participants tolerated well the amounts of folate-rich foods and vegetables offered. Based on the many studies cited in the present work, we observed the use of folic acid supplementation, folate fortification, vegetable consumption, etc., however, we did not see many studies with low folate doses (< 400 mcg / day) with good results. Effective changes in dietary habits that incorporate adequate amounts of nutrients such as folate, need involve an educational processes that require investments and efforts from the government. According to the literature consulted the intake of dietary folate is insufficient to supply EAR, so in countries where folic acid fortification is not available, an insufficient intake of folic acid is more pronounced (Shi, Caprau et al., 2003). Thus, greater inadequate folate consumption would contribute to more detrimental health outcomes in individuals with the MTHFR C677T polymorphism.
Discussion: The limitations and lack of group control of the study can be found in the last paragraph of discussion section on lines 479-485 or below:
The sample size was the limitation of the study. However, even so we still yet observed a dietary response with the highest dose of food folate, even under the influence of the MTHFR C677T polymorphism. Thus, the results of this study demonstrate the importance of considering the individuality information when establishing strategic nutritional consumption goals for specific biological responses. The control group was not used in this study because the researchers wanted to investigate which dose of folate in the natural diet had benefits for obese adult women with the C677T polymorphism.
The reason for being only two groups with two doses of folate was to women was analyzed if a lower dose had already been increased or if it was a higher dose would be better for this type of genetic profile (MTHFR gene C677T polymorphism) via food Natural and comedy of the strength for advanced in the manufacturing and incorporated in Brazil.
Our best regards.
We're hope that the responses would be satisfactory to you, but if they're not, we're available for any further questions or doubts.
